# Distribution Oblivious, Risk-Aware Algorithms for Multi-Armed Bandits with Unbounded Rewards

**Anmol Kagrecha**
IIT Bombay
akagrecha@gmail.com

**Jayakrishnan Nair**
IIT Bombay
jayakrishnan.nair@ee.iitb.ac.in

**Krishna Jagannathan**
IIT Madras
krishnaj@ee.iitm.ac.in

## Abstract

Classical multi-armed bandit problems use the expected value of an arm as a metric to evaluate its goodness. However, the expected value is a risk-neutral metric. In many applications like finance, one is interested in balancing the expected return of an arm (or portfolio) with the risk associated with that return. In this paper, we consider the problem of selecting the arm that optimizes a linear combination of the expected reward and the associated Conditional Value at Risk (CVaR) in a fixed budget best-arm identification framework. We allow the reward distributions to be unbounded or even heavy-tailed. For this problem, our goal is to devise algorithms that are entirely *distribution oblivious*, i.e., the algorithm is not aware of any information on the reward distributions, including bounds on the moments/tails, or the suboptimality gaps across arms.

In this paper, we provide a class of such algorithms with provable upper bounds on the probability of incorrect identification. In the process, we develop a novel estimator for the CVaR of unbounded (including heavy-tailed) distributions and prove a concentration inequality for the same, which could be of independent interest. We also compare the error bounds for our distribution oblivious algorithms with those corresponding to standard non-oblivious algorithms. Finally, numerical experiments reveal that our algorithms perform competitively when compared with non-oblivious algorithms, suggesting that distribution obliviousness can be realised in practice without incurring a significant loss of performance.

## 1 Introduction

The multi-armed bandit (MAB) problem is fundamental in online learning, where an optimal option needs to be identified among a pool of available options. Each option (or arm) generates a random reward/cost when chosen (or pulled) from an underlying unknown distribution, and the goal is to quickly identify the optimal arm by exploring all possibilities.

Classically, MAB formulations consider reward distributions with bounded support, typically $[0, 1]$. Moreover, the support is assumed to be known beforehand, and this knowledge is baked into the algorithm. However, in many applications, it is more natural to not assume bounded support for the reward distributions, either because the distributions are themselves unbounded, or because a bound on the support is not known *a priori*. There is some literature on MAB formulations with (potentially) unbounded rewards; see, for example, Bubeck et al. [2013], Vakili et al. [2013]. Typically, in these papers, the assumption of a known bound on the support of the reward distributions is replaced with the assumption that certain bounds on the moments/tails of the reward distributions are known.[1] However, such access to prior information is not always practical, and goes against the spirit of *online* learning. This motivates the design and analysis of algorithms for the MAB problem that are *distribution oblivious*, i.e., algorithms that have zero prior knowledge about the reward distributions.

Furthermore, the typical metric used to quantify the goodness of an arm in the MAB framework is its expected return, which is a risk-neutral metric. In some applications, particularly in finance, one is interested in balancing the expected return of an arm with the risk associated with that arm. This is particularly relevant when the underlying reward distributions are unbounded, even heavy-tailed, as is found to be the case with portfolio returns in finance Bradley and Taqqu [2003]. In these settings, there is a non-trivial probability of a 'catastrophic' outcome, which motivates a risk-aware approach to optimal arm selection.

In this paper, we seek to address the two issues described above. Specifically, we consider the problem of identifying the arm that optimizes a linear combination of the reward and the Conditional Value at Risk (CVaR) in a fixed budget (pure exploration) MAB framework. The CVaR is a classical metric used to capture the risk associated with an option/portfolio Artzner et al. [1999]. We make very mild assumptions on the reward distributions (the existence of a $(1 + \epsilon)$th moment for some $\epsilon > 0$), allowing for unbounded support and even heavy tails. In this setting, our goal is to design algorithms that are entirely distribution oblivious.

The main contribution of this paper is the design and analysis of distribution oblivious algorithms for the risk-aware best arm identification problem described above. These algorithms are based on truncation-based estimators for the mean and CVaR, where the truncation parameters are scaled suitably as the algorithm runs. We prove upper bounds on the probability of incorrect arm identification for these algorithms that have the form $O(\exp(-\gamma T^{1-q}))$, where $T$ is the budget of arm pulls, $\gamma > 0$ is a constant that depends on the arm distributions, and $q \in (0, 1)$ is an algorithm parameter. Note the slower-than-exponential decay in the probability of erroneous arm identification with respect to $T$. This is a consequence of the distribution obliviousness of the proposed algorithms. Indeed, in the non-oblivious setting, it is easy to develop algorithms with an $O(\exp(-\gamma' T))$ probability of error. Moreover, numerical experiments show that the proposed distribution oblivious algorithms perform competitively when compared with standard non-oblivious algorithms. This suggests that distribution obliviousness can be realised in practice without incurring a significant performance hit. Finally, we note that the truncation-based CVaR estimator used in our algorithms is novel, and the concentration inequality we prove for this estimator may be of independent interest.

The remainder of this paper is organized as follows. A brief survey of the related literature is provided below, followed by some preliminaries. Our CVaR concentration results are presented in Section 2, and our distribution oblivious algorithms for risk-aware best arm identification are proposed and analysed in Section 3. Numerical experiments are presented in Section 4, and we conclude in Section 5. Throughout the paper, references to the appendix (primarily for proofs) point to the 'additional material' document uploaded separately.

## 1.1  Related Literature

There is a considerable body of literature on the multi-armed bandit problem. We refer the reader to Bubeck and Cesa-Bianchi [2012], Lattimore and Szepesvári [2018] for a comprehensive review. Here, we restrict ourselves to papers that consider (i) unbounded reward distributions, and (ii) risk-aware arm selection.

The papers that consider MAB problems with (potentially) heavy-tailed reward distributions include: Bubeck et al. [2013], Vakili et al. [2013], Carpentier and Valko [2014], which consider the regret minimization framework, and Yu et al. [2018], which considers the pure exploration framework. All the above papers take the expected return of an arm to be its goodness metric. Bubeck et al. [2013], Vakili et al. [2013] assume prior knowledge of moment bounds and/or the suboptimality gaps. Carpentier and Valko [2014] assumes that the arms belong to parameterized family of distributions satisfying a second order Pareto condition. Yu et al. [2018] does analyse one distribution oblivious algorithm (see Theorem 2 in the paper), though the performance guarantee derived there is much weaker than the ones proved here; we elaborate on this in Section 3.

There has been some recent interest in risk-aware multi-armed bandit problems. Sani et al. [2012] considers the setting of optimizing a linear combination of mean and variance in the regret minimization framework. In the pure exploration setting, VaR-optimization has been considered in David and Shimkin [2016], David et al. [2018]. However, the CVaR is a more preferable metric because it is a coherent risk measure (unlike the VaR); see Artzner et al. [1999]. Strong concentration results for VaR are available without any assumptions on the tail of the distribution Kolla et al. [2019], whereas

concentration results for CVaR are more difficult to obtain. Assuming bounded rewards, Galichet et al. [2013] have considered the problem of CVaR-optimization. In a recent paper, Prashanth et al. [2019] consider CVaR optimization with heavy tailed distributions, but they assume prior knowledge of moment bounds. None of the above papers consider the problem of risk-aware arm selection in a distribution oblivious fashion, as is done here.

## 1.2 Preliminaries

Here, we define the Value at Risk (VaR) and the Conditional Value at Risk (CVaR), and state the assumptions we make in this paper on the arm distributions.

For a random variable $X$, given a prescribed confidence level $\alpha \in (0, 1)$, the *Value at Risk* (VaR) is defined as $v_\alpha(X) = \inf(\xi : \mathbb{P}(X \leq \xi) \geq \alpha)$. If $X$ denotes the loss associated with a portfolio, $v_\alpha(X)$ can be interpreted as the worst case loss corresponding to the confidence level $\alpha$. The *Conditional Value at Risk* (CVaR) of $X$ at confidence level $\alpha \in (0, 1)$ is defined as

$$c_\alpha(X) = v_\alpha(X) + \frac{1}{1 - \alpha} \mathbb{E}[X - v_\alpha(X)]^+,$$

where $[z]^+ = \max(0, z)$. Both VaR and CVaR are used extensively in the finance community as measures of risk, through the CVaR is often preferred as mentioned above. Typically, the confidence level $\alpha$ is chosen between 0.95 and 0.99. Throughout this paper, we use the CVaR as a measure of the risk associated with an arm. We define $\beta := 1 - \alpha$. For the special case where $X$ is continuous with a cumulative distribution function (CDF) $F_X$ that is strictly increasing over its support, $v_\alpha(X) = F_X^{-1}(\alpha)$. In this case, the CVaR can also be written as $c_\alpha(X) = \mathbb{E}[X|X \geq v_\alpha(X)]$. Going back to our portfolio loss analogy, $c_\alpha(X)$ can, in this case, be interpreted as the expected loss conditioned on the 'bad event' that the loss exceeds the VaR.

For our analysis, we assume that the arm distributions satisfy the following condition: A random variable $X$ satisfies condition **C1** if there exists $p > 1$ and $B < \infty$ such that $\mathbb{E}[|X|^p] < B$. Note that **C1** is only mildly more restrictive than assuming the well-posedness of the MAB problem, which requires $\mathbb{E}[|X|] < \infty$.[2] In particular, all light-tailed distributions and most heavy-tailed distributions used and observed in practice satisfy **C1**.

## 2  CVaR Concentration

In this section, we derive a concentration inequality for an estimator of the CVaR corresponding to a distribution with unbounded support. The key feature of this concentration inequality is that it makes very mild assumptions on the tail of the distribution; specifically, our concentration result applies even to heavy-tailed distributions (unlike prior results in the literature, that assume a bounded distribution Wang and Gao [2010], or a subgaussian/subexponential tail Kolla et al. [2019]). This CVaR concentration result (Theorem 2 below), while of independent interest, will be invoked in Section 3 to prove guarantees on our algorithms for the risk-aware multi-armed bandit problem.

Assume that $\{X_i\}_{i=1}^n$ are $n$ i.i.d. samples distributed as the random variable $X$. Let $\{X_{[i]}\}_{i=1}^n$ denote the order statistics of $\{X_i\}_{i=1}^n$ i.e., $X_{[1]} \geq X_{[2]} \cdots \geq X_{[n]}$. Recall that the classical estimator for $c_\alpha(X)$ given the samples $\{X_i\}_{i=1}^n$ is

$$\hat{c}_{n,\alpha}(X) = X_{[\lceil n\beta \rceil]} + \frac{1}{n\beta} \sum_{i=1}^{\lfloor n\beta \rfloor} (X_{[i]} - X_{[\lceil n\beta \rceil]}).$$

We begin by proving a concentration inequality for $\hat{c}_{n,\alpha}(X)$ for the special case when $X$ is bounded.

**Theorem 1.** *For $b > 0$, suppose that $X$ satisfies supp($X$) $\subseteq [-b, b]$. Then for any $\varepsilon > 0$,*

$$\Pr\left(|\hat{c}_{n,\alpha}(X) - c_\alpha(X)| \geq \varepsilon\right) \leq 6\exp\left(-\frac{n\beta(\varepsilon/b)^2}{34 + 4.4\varepsilon/b}\right).$$

Theorem 1 is a refinement of the CVaR concentration inequality for bounded distributions in Wang and Gao [2010]. The proof can be found in Appendix A.

We now use Theorem 1 to develop a CVaR concentration inequality for unbounded (potentially heavy-tailed) distributions. In particular, our concentration inequality applies to the following truncation-based estimator. For $b > 0$, define

$$X_i^{(b)} = \min(\max(-b, X_i), b).$$

Note that $X_i^{(b)}$ is simply the projection of $X_i$ onto the interval $[-b, b]$. Let $\{X_{[i]}^{(b)}\}_{i=1}^n$ denote the order statistics of truncated samples $\{X_i^{(b)}\}_{i=1}^n$. Our estimator $\hat{c}_{n,\alpha}^{(b)}(X)$ for $c_\alpha(X)$ is simply the empirical CVaR estimator for $X^{(b)} := \min(\max(-b, X), b)$, i.e.,

$$\hat{c}_{n,\alpha}^{(b)}(X) = \hat{c}_{n,\alpha}(X^{(b)}) = X_{[\lceil n\beta \rceil]}^{(b)} + \frac{1}{n\beta} \sum_{i=1}^{\lfloor n\beta \rfloor} (X_{[i]}^{(b)} - X_{[\lceil n\beta \rceil]}^{(b)}).$$

Note that the nature of truncation performed here is different from that in the conventional truncation-based mean estimators (see, for example, Bubeck et al. [2013]), where samples with an absolute value greater than $b$ are set to zero. In contrast, our estimator projects these samples to the interval $[-b, b]$. This difference plays an important role in establishing the concentration properties of the estimator.

We are now ready to state our main result, which shows that the truncation-based estimator $\hat{c}_{n,\alpha}^{(b)}(X)$ works well when the truncation parameter $b$ is large enough.

**Theorem 2.** *Suppose that $\{X_i\}_{i=1}^n$ are i.i.d. samples distributed as $X$, where $X$ satisfies condition* **C1**. *Given $\Delta > 0$,*

$$\Pr\left(|c_\alpha(X) - \hat{c}_{n,\alpha}^{(b)}(X)| \geq \Delta\right) \leq 6\exp\left(-n(1-\alpha)\frac{\Delta^2}{154b^2}\right) \tag{1}$$

$$\text{for } b > \max\left(\frac{\Delta}{2}, |v_\alpha(X)|, \left[\frac{2B}{\Delta(1-\alpha)}\right]^{\frac{1}{p-1}}\right). \tag{2}$$

The proof of Theorem 2 can be found in Appendix B. The key feature of truncation-based estimators like the one proposed here for the CVaR is that they enable a parameterized bias-variance trade-off. While the truncation of the data itself adds a bias to the estimator, the boundedness of the (truncated) data limits the variability of the estimator. Indeed, the condition that $b > \left[\frac{2B}{\Delta(1-\alpha)}\right]^{\frac{1}{p-1}}$ in the statement of Theorem 2 ensures that the estimator bias induced by the truncation is at most $\Delta/2$.

In practice, one might not know the values of $v_\alpha(X)$, $B$, $p$ or even $\Delta$ (as is the case in MAB problems), so ensuring that the lower bound on $b$ is satisfied is problematic.[3] The natural strategy to follow then is to set the truncation parameter as an increasing function of the number of data samples $n$, which ensures that (2) holds for large enough $n$. Moreover, it is clear from (1) that for the estimation error to (be guaranteed to) decay with $n$, $b^2$ can grow at most linearly in $n$. Indeed, for our bandit algorithms, we set $b = n^q$, where $q \in (0, 1/2)$.

Finally, it is tempting to set $b$ in a *data-driven* manner, i.e., to estimate the VaR, moment bounds and so on from the data, and set $b$ large enough so that (2) holds with high probability. The issue however is that $b$ then becomes a (data-dependent) random variable, and proving concentration results with such data-dependent truncation is much harder.

## 3    Risk-aware, distribution oblivious algorithms for MAB

In this section, we formulate the problem of best arm identification in a risk-aware fashion, propose algorithms, and prove performance guarantees for these algorithms.

Consider a multi-armed bandit problem with $K$ arms, labeled $1, 2, \cdots, K$. The loss (or cost) associated with arm $i$ is distributed as $X(i)$, where it is assumed that there exists $p > 1$ and $B < \infty$

such that $\mathbb{E}\left[|X(i)|^p\right] < B$ for all $i$.[4] Each time an arm $i$ is pulled, an independent sample distributed as $X(i)$ is observed. Given a fixed budget of $T$ arm pulls in total, our goal is to identify the arm that minimizes $\xi_1 \mathbb{E}\left[X(i)\right] + \xi_2 c_\alpha(X(i))$, where $\xi_1$ and $\xi_2$ are non-negative (and given) weights. $(\xi_1, \xi_2) = (1, 0)$ corresponds to the classical mean minimization problem Audibert et al. [2010], Yu et al. [2018], whereas $(\xi_1, \xi_2) = (0, 1)$ corresponds to a pure CVaR minimization Galichet et al. [2013], Prashanth et al. [2019]. Optimization of a linear combination of the mean and CVaR has been considered before in the context of portfolio optimization in the finance community, but not, to the best of our knowledge, in the MAB framework. The performance metric we consider is the probability of incorrect arm identification.

We also assume that there is a unique optimal arm. This is purely for simplicity in expressing our performance guarantees; it is straightforward to extend these to the setting where there are multiple optimal arms. Let the ordered sub-optimality gaps for the metric $\xi_1 \mathbb{E}\left[X(\cdot)\right] + \xi_2 c_\alpha(X(\cdot))$ be denoted by $\Delta[2], \cdots, \Delta[K]$; here, $0 < \Delta[2] \leq \cdots \leq \Delta[K]$.

Finally, recall that we consider an entirely distribution oblivious environment. In other words, the algorithm does not have any prior information about the arm distributions, including the values of $p$ and $B$. This is in contrast with the most of the literature on MAB problems, where information about the support of the arm distributions, bounds on their moments and/or sub-optimality gaps are baked into the algorithms.[5]

## 3.1 Algorithms

We estimate the performance of each arm as follows. Suppose that arm $i$ has been pulled $n$ times, and we observe samples $X_1^i, X_2^i, \cdots, X_n^i$. We use the following truncated empirical estimator (see Bickel [1965], Bubeck et al. [2013]) for the mean value associated with the arm:

$$\hat{\mu}_n^\dagger(i) := \frac{\sum_{j=1}^n X_j^i \mathbb{1}\left\{|X_j^i| \leq b_m(n)\right\}}{n},$$

where $b_m(n) = n^{q_m}$ for $q_m \in (0, 1)$. Note that we are growing the truncation parameter $b_m$ sublinearly in $n$. Our estimator for the CVaR associated with arm $i$ is the one developed in Section 2, i.e.,

$$\hat{c}_{n,\alpha}^\dagger = \hat{c}_{n,\alpha}^{(b_c(n))},$$

where $b_c(n) = n^{q_c}$ for $q_c \in (0, 1/2)$.

Our algorithms are of *successive rejects* (SR) type Audibert et al. [2010]. They are parameterized by non-negative integers $n_1 \leq n_2 \leq \cdots \leq n_{K-1}$ satisfying $\sum_{i=1}^{K-2} n_i + 2n_{K-1} \leq T$. The algorithm proceeds in $K-1$ phases, with one arm being rejected from further consideration at the end of each phase. In phase $i$, the $K-1+i$ arms under consideration are pulled $n_i - n_{i-1}$ times, after which the arm with the worst (estimated) performance is rejected. This is formally expressed in Algorithm 1. The classical successive rejects algorithm in Audibert et al. [2010] used $n_k \propto \frac{T-K}{K+1-k}$. Another special case is *uniform exploration* (UE), where $n_1 = n_2 = \cdots n_{K-1} = \lfloor T/K \rfloor$. As the name suggests, under uniform exploration, all arms are pulled an equal number of times, after which the arm with the best estimate is selected.

If the time horizon $T$ is not known a priori, an any-time variant can be constructed by modifying the UE algorithm using the doubling trick Lattimore and Szepesvári [2018]. The classical SR algorithm, however, cannot be made any-time since the duration of each phase depends on the horizon $T$.

## 3.2 Performance evaluation

We now state upper bounds on the probability of incorrect arm identification under the successive rejects and uniform exploration algorithms. However, our bounding techniques easily extend to the complete class of generalized successive rejects algorithms described in Algorithm 1.

**Algorithm 1** Generalized successive rejects algorithm
---
    **procedure** GSR($T, K, \{n_1, \cdots, n_{K-1}\}$)
        $A_1 \leftarrow \{1, \cdots, K\}$
        $n_0 \leftarrow 0$
        **for** $k = 1$ to $K - 1$ **do**
            **for** $i \in A_k$ **do**
                Sample arm $i$ for $n_k - n_{k-1}$ rounds.
            **end for**
            $\tilde{a} \leftarrow \arg\max_{i \in A_k} \xi_1 \mu^{\dagger}_{n_k}(i) + \xi_2 \hat{c}^{\dagger}_{n_k, \alpha}(i)$
            $A_{k+1} \leftarrow A_k \setminus \tilde{a}$
        **end for**
        Output unique element of $A_K$
    **end procedure**
---

**Theorem 3.** *Suppose that the arm distributions satisfy the condition **C1**. Under the uniform exploration algorithm, the probability of incorrect arm identification $p_e$ is bounded as*

$$p_e \leq 2K exp\Big(-(T/K)^{1-q_m}\frac{\Delta[2]}{16\xi_1}\Big) + 6K exp\Big(-(T/K)^{1-2q_c}\frac{\beta\Delta[2]^2}{2464\xi_2^2}\Big)$$

*for $T > Kn^*$, where*

$$n^* = \max\left(\Big(\frac{12\xi_1 B}{\Delta[2]}\Big)^{\frac{1}{q_m \min(p-1,1)}}, \Big(\frac{8\xi_2 B}{\beta\Delta[2]}\Big)^{\frac{1}{q_c(p-1)}}, \Big(\frac{B}{min(\alpha,\beta)}\Big)^{\frac{1}{q_c p}}, \Big(\frac{\Delta[2]}{8\xi_2}\Big)^{\frac{1}{q_c}}\right).$$

The proof of Theorem 3 can be found in Appendix C. Here, we highlight the main takeaways from this result.

First, note that the probability of error (incorrect arm identification) decays to zero as $T \to \infty$. However, *the decay is slower than exponential in $T$*; taking $q_m = q$, $q_c = q/2$ for $q \in (0,1)$, the probability of error is $O(\exp(-\gamma T^{1-q}))$ for a positive constant $\gamma$. This slower-than-exponential bound is a consequence of the distribution obliviousness of the algorithm. In technical terms, this results from having to set the truncation parameters $b_m$ and $b_c$ for each arm as increasing functions of the horizon $T$. Indeed, as we show in Section 3.3, if $B$, $p$, and $\Delta[2]$ are known to the algorithm (as is often assumed in the literature), then it is possible to achieve an exponential decay of the probability of error with $T$; in this case, it is possible to simply set the truncation parameters as static constants (that do not depend on $T$).

Second, our upper bounds only hold when $T$ is larger than a certain threshold. This is again a consequence of distribution obliviousness—the concentration inequalities on our truncated estimators are only valid when the truncation interval is wide enough. This is required in order to limit the bias of these estimators. As a consequence, our performance guarantees only kick in once the horizon length is large enough to ensure that this condition is met. As expected, in the non-oblivious setting, this limitation does not arise, since the truncation parameters can be statically set to be large enough to limit the bias (see Section 3.3).

Third, there is a natural tension between the bound for the probability of error and the threshold on $T$ beyond which they are applicable, with respect to the choice of truncation parameters $q_m$ and $q_c$. In particular, the upper bound on $p_e$ decays fastest with respect to $T$ when $q_m, q_c \approx 0$. However, choosing $q_m, q_c$ to be small would make the threshold on the horizon to be large, since the bias of our estimators would decay slower with respect to $T$. Intuitively, smaller values of $q_m, q_c$ limit the variance of our estimators (which is reflected in the bound for $p_e$) at the expense of a greater bias (which is reflected in the threshold on $T$), whereas larger values $q_m, q_c$ limit the bias at the expense of increased variance. We comment on the best choice of these parameters as suggested by numerical experimentation in Section 4.

Finally, we note that the bound on the probability of error in Theorem 4 is stronger than the power law bound corresponding to the distribution oblivious algorithm for the mean metric analysed in Yu et al. [2018]. The latter uses the standard (non-truncated) empirical mean estimator, which has weaker concentration properties compared to the truncated empirical mean estimator used here.

Next, we consider the successive rejects algorithm. Let $\overline{\log}(K) := 1/2 + \sum_{i=2}^{K} 1/i$.

**Theorem 4.** *Let the arms satisfy the condition **C1**. The probability of incorrect arm identification for the successive rejects algorithm is bounded as follows.*

$$p_e \leq \sum_{i=2}^{K} (K+1-i)2exp\left( -\frac{1}{16\xi_1}\left(\frac{T-K}{\overline{\log}(K)}\right)^{1-q_m}\frac{\Delta[i]}{i^{1-q_m}} \right)$$

$$+ \sum_{i=2}^{K} (K+1-i)6exp\left( -\frac{\beta}{2464\xi_2^2}\left(\frac{T-K}{\overline{\log}(K)}\right)^{1-2q_c}\frac{\Delta[i]^2}{i^{1-2q_c}} \right)$$

*for $T > K + K\overline{\log}(K)n^*$, where*

$$n^* = \max\left( \left(\frac{12\xi_1 B}{\Delta[2]}\right)^{\frac{1}{q_m \min(p-1,1)}}, \left(\frac{8\xi_2 B}{\beta\Delta[2]}\right)^{\frac{1}{q_c(p-1)}}, \left(\frac{B}{\min(\alpha,\beta)}\right)^{\frac{1}{q_c p}}, \left(\frac{\Delta[2]}{8\xi_2}\right)^{\frac{1}{q_c}} \right).$$

Structurally, our results for the successive rejects algorithm are similar to those for uniform exploration. Indeed, taking $q_m = q$, $q_c = q/2$ for $q \in (0,1)$, the probability of error remains $O(exp(-\gamma T^{1-q}))$ for a different positive constant $\gamma$. So our conclusions from Theorem 3, including the bias-variance trade-off in setting the truncation parameters $q_m$ and $q_c$, apply to Theorem 4 as well. Intuitively, one would expect the successive rejects algorithm to perform better when the arms are well separated, whereas uniform exploration would work well when all sub-optimal arms are nearly identically separated from the optimal arm.

### 3.3 The non-oblivious setting

Finally, we consider the non-oblivious setting, where the algorithm knows $p$, $B$ and $\Delta[2]$ (or a lower bound on $\Delta[2]$). This is the setting that is effectively considered in the bulk of the literature on MAB algorithms. In this case, we show that it is possible to set the algorithm parameters (specifically, the truncation parameters) so that we achieve an exponential decay of the probability of error with $T$. Moreover, unlike our results for the distribution oblivious case, there is no lower bound on $T$ beyond which the bounds on the probability of error apply.

In particular, we set truncation threshold for the mean estimator as $b_m = \left(\frac{12B\xi_1}{\Delta[2]}\right)^{\frac{1}{\min(1,p-1)}}$ and the truncation threshold for the CVaR estimator as $b_c = \max\left( \left(\frac{8\xi_2 B}{\beta\Delta[2]}\right)^{\frac{1}{p-1}}, \left(\frac{B}{\min(\alpha,\beta)}\right)^{\frac{1}{p}} \right)$. It can be shown that this would ensure an exponentially decaying (in $T$) probability of error for uniform exploration as well as successive rejects (see Appendix D).

In conclusion, the results in this section show that one can indeed devise algorithms for risk-aware best arm identification in an entirely distribution oblivious manner. However, the performance guarantees we obtain are not as strong as those that can be obtained for non-oblivious algorithms; this is of course what one would expect.

## 4 Numerical Experiments

In this section, we evaluate the performance of the proposed algorithms via simulations, by making the comparison with (more conventional) non-oblivious algorithms. Due to space constraints, we restrict ourselves to successive rejects (SR) algorithms, and two specific objectives: (i) mean minimization, i.e., $(\xi_1, \xi_2) = (0,1)$, and (ii) CVaR minimization, i.e., $(\xi_1, \xi_2) = (1,0)$. In each of the experiments below, the probability of error is computed by averaging over 50000 runs at each sampled $T$. For the mean minimization problem, our results are compared to the non-oblivious truncation based SR algorithm from Yu et al. [2018] where $b_m = (6Bp/\Delta[2])^{1/\min(1,p-1)}$. The proposed oblivious SR algorithm for CVaR minimization is compared to a non-oblivious truncation based SR algorithm with $b_c = (4B/(\Delta[2]\beta))^{1/(p-1)}$, which ensures an exponential decay in $T$ of the error probability upper bound. For CVaR minimization problems, the confidence level is set to 0.95.

Consider the case when all the arms are light-tailed. In particular, for mean minimization we consider the following MAB problem instance: there are 10 arms, exponentially distributed, the optimal

having mean loss 0.97, and the remaining having mean loss 1. For CVaR minimization, consider the following MAB problem instance: there are 10 arms, exponentially distributed, the optimal having a CVaR 2.85, and the remaining having a CVaR 3.00. The non-oblivious algorithms also know that $p = 2.0$, $B = 2.0$ for both settings as well as $\Delta[2] = 0.03$ for mean minimization and $\Delta[2] = 0.15$ for CVaR minimization. Oblivious truncation parameters for both the settings are growing as $n^{0.3}$. As can be seen in Figure 1a and Figure 1b, the oblivious and the non-oblivious algorithms perform equally well. The estimators concentrate around the true values easily when the underlying distributions are light-tailed.

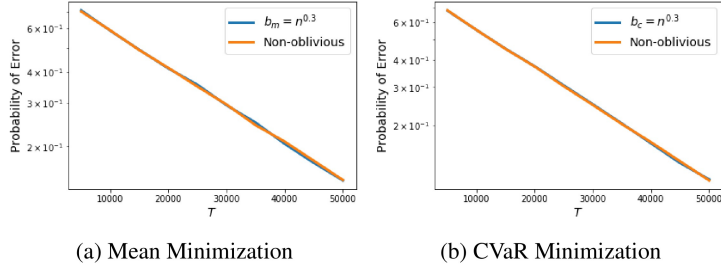

(a) Mean Minimization          (b) CVaR Minimization

Figure 1: Exponentially Distributed Arms

Now, consider the case when all the arms are heavy-tailed. For mean minimization we consider the following MAB problem instance: there are 10 arms, distributed according to the lomax distribution (scale parameter = 1.8), the optimal arm having mean loss 0.9, and the remaining arms having mean loss 1. The parameters available to the non-oblivious algorithm are: $p = 1.7$, $B = 10.8$ and $\Delta[2] = 0.1$. For CVaR minimization, consider the following MAB problem instance: there are 10 arms, distributed according to lomax distribution (scale parameter = 2.0), the optimal having a CVaR 2.55, and the remaining having a CVaR 3.00. Here, the parameters for the non-oblivious algorithm are: $p = 1.9$, $B = 0.057$ and $\Delta[2] = 0.45$. Note that like the previous case, the truncation parameters grow as $n^{0.3}$.

As can be seen in Figure 2a and Figure 2b, the non-oblivious algorithms show a visible hit in performance when compared to the oblivious algorithms. The truncation parameters for the non-oblivious algorithms are much larger compared to the truncation parameters of the oblivious algorithms. The smaller truncation introduces a bias in estimation but reduces variability of the estimator considerably. Despite having a smaller bias, the non-oblivious estimators suffer due to the variability induced by the heavy-tailed cost distributions.

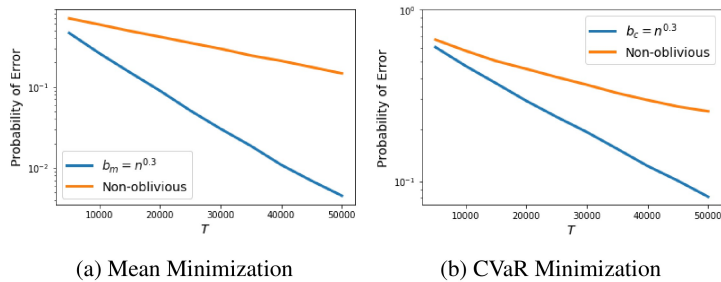

(a) Mean Minimization          (b) CVaR Minimization

Figure 2: Lomax Distributed Arms

Finally, we consider the case where we have both light-tailed and heavy-tailed arms. First, consider the case where the optimal arm is light tailed. The mean minimization MAB problem instance is as follows: there are 10 arms, five distributed according to lomax distribution (scale parameter=1.8), and five distributed exponentially, the optimal having mean loss 0.9, and the remaining having mean loss 1.0. The parameters provided to the non-oblivious algorithm are: $p = 1.7$, $B = 10.8$ and $\Delta[2] = 0.1$. For CVaR minimization, we set the confidence value to be 0.95 and consider the following MAB problem instance: there are 10 arms, five distributed according to lomax distribution (scale parameter=2.0), and five distributed exponentially, the optimal arm having a CVaR 2.55, and

the remaining arms having a CVaR 3.00. The non-oblivious algorithm knows the following: $p = 1.9$, $B = 0.057$ and $\Delta[2] = 0.45$.

It can be observed in Figure 3a and Figure 3b that growing the truncation parameter as $n^{0.3}$ leads to a poor performance of the oblivious algorithms. Small truncation parameters lead to an underestimation of the mean, which affects the heavy tailed arms much more than the light tailed arms and causes the algorithm to misidentify the sub-optimal heavy tailed arms as optimal. When the truncation growth is fast enough, for example, $n^{0.8}$ for mean minimization and $n^{0.45}$ for CVaR minimization, the performance is as good as the non-oblivious algorithms, which use estimators with large truncation values.

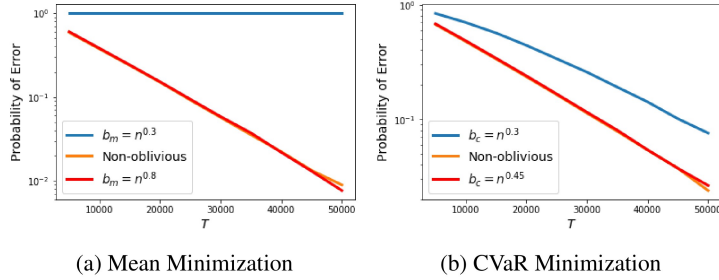

(a) Mean Minimization          (b) CVaR Minimization

Figure 3: Mixture of Exponential and Lomax Arms with an Exponential Arm Optimal

The case where the optimal arm is heavy-tailed is less interesting. Here, the greater underestimation of the true value for the heavy tailed arms actually aids the proposed distribution oblivious algorithms. As a result, the distribution oblivious algorithms perform very well; the results are omitted due to space constraints.

## 5    Concluding Remarks

In this paper, we consider the problem of risk-aware best arm selection in a pure exploration MAB framework. A key feature of our algorithms is *distribution obliviousness*; the algorithms have no prior knowledge about the arm distributions. This is in contrast with most algorithms in the literature for MAB problems, which assume prior knowledge of the support, moment bounds, or bounds on the sub-optimality gaps. The proposed algorithms come with analytical performance guarantees, and also seem to perform well in practice.

This paper motivates future work along several directions. First, our numerical experiments suggest that our upper bounds on the probability of error for the distribution oblivious algorithms are rather loose. Tighter performance bounds, which would in turn require tighter concentration bounds for truncation-based estimators, are worth exploring. More importantly, fundamental lower bounds on the performance of any algorithm need to be devised for the distribution oblivious setting. Currently, available lower bounds (see Audibert et al. [2010]) on the error probability for best arm identification do not take into account the information available to the algorithm, and only capture risk-neutral arm selection. Finally, it is also interesting to explore distribution oblivious algorithms in the regret minimization framework, as well as the PAC framework.

## Acknowledgements

The first author is grateful to Google for a generous travel award to enable him to present this work at NeurIPS 2019.

## Footnotes

[1]Additionally, many algorithms require knowledge of a lower bound on the sub-optimality gap between arms.

[2]Of course, our distribution oblivious algorithms do not know the values of $p, B$ for any of the arms.

[3]We note here that $|v_\alpha(X)|$ can be upper bounded in terms of $p$ and $B$ as follows: $|v_\alpha(X)| \leq \left(\frac{B}{\min(\alpha, \beta)}\right)^{\frac{1}{p}}$ (see Appendix C.2). Thus, $b > \left(\frac{B}{\min(\alpha, \beta)}\right)^{\frac{1}{p}}$ implies $b > |v_\alpha(X)|$.

[4]We pose the problem as (risk-aware) loss minimization, which is of course equivalent to (risk-aware) reward maximization.

[5]While the algorithms we propose are distribution oblivious, their performance guarantees will of course depend on the arm distributions.

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
