[Supplementary Material]

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

# A  CVaR Concentration for Bounded Random Variables (Proof of Theorem 1)

We state two concentration inequalities that will be used repeatedly in the proof of Theorem 1.

**Bernstein's Inequality**

Let $X_i$ be IID samples of a random variable $X$ with mean $\mu$. If $|X| \leq b$ almost surely, then for any $\epsilon > 0$,

$$\mathbb{P}\left( \left| \frac{\sum_{i=1}^n X_i}{n} - \mu \right| > \epsilon \right) \leq 2\exp\left( - \frac{n\epsilon^2}{2\mathbb{E}[X^2] + 2b\epsilon/3} \right)$$

**Chernoff Bound for Bernoulli Experiment**

Let $X_1, ..., X_n$ be independent Bernoulli experiments, $\mathbb{P}(X_i = 1) = p_i$. Set $X = \sum_{i=1}^n X_i$, $\mu = \mathbb{E}[X]$. Then for every $0 < \delta < 1$,

$$P(X \leq (1-\delta)\mu) \leq \exp(-\frac{\mu\delta^2}{2}),$$

for every $\delta > 0$,

$$P(X \geq (1+\delta)\mu) \leq \exp(-\frac{\mu\delta^2}{2+\delta}),$$

and in particular, for every $0 < \delta < 2$,

$$P(X \geq (1+\delta)\mu) \leq \exp(-\frac{\mu\delta^2}{4})$$

Theorem 1 follows from the following statement. Let $X$ be any random variable with $\text{supp}(X) \subseteq [-b, b]$. Then for any $\varepsilon \geq 0$,

$$\mathbb{P}(\hat{c}_{n,\alpha}(X) \leq c_\alpha(X) - \varepsilon) \leq 3\exp\left( - (1-\alpha)n\frac{(\varepsilon/b)^2}{9 + 1.6\varepsilon/b} \right) \tag{3a}$$

$$\mathbb{P}(\hat{c}_{n,\alpha}(X) \geq c_\alpha(X) + \varepsilon) \leq 3\exp\left( - n(1-\alpha)\frac{(\varepsilon/b)^2}{34 + 4.4\varepsilon/b} \right) \tag{3b}$$

## A.1  Proof of 3a

We're going to use the following lemma from Wang and Gao [2010].

**Lemma 1.** *Let $X_{[i]}$ be the decreasing order statistics of $X_i$; then $f(k) = \frac{1}{k}\sum_{i=1}^k X_{[i]}$, $1 \leq k \leq n$, is decreasing and the following two inequalities hold:*

$$\frac{1}{n\beta}\sum_{i=1}^{\lfloor n\beta \rfloor} X_{[i]} \leq \hat{c}_{n,\alpha}(X) \leq \frac{1}{n\beta}\sum_{i=1}^{\lceil n\beta \rceil} X_{[i]} \tag{4a}$$

$$f(\lceil n\beta \rceil) \leq \hat{c}_{n,\alpha}(X) \leq f(\lfloor n\beta \rfloor) \tag{4b}$$

**1.** $\varepsilon > 2b$

$$\mathbb{P}(\hat{c}_{n,\alpha}(X) \leq c_\alpha(X) - \varepsilon) = 0$$

Both $c_\alpha(X) \in [-b, b]$ and $\hat{c}_{n,\alpha}(X) \in [-b, b]$. Therefore, the difference can't be larger than $2b$.

**2.** $\varepsilon \in [0, 2b]$

We'll condition the probability above on a random variable $K_{n,\beta}$ which is defined as $K_{n,\beta} = \max\{i : X_{[i]} \in [v_\alpha(X), b]\}$. Note that $v_\alpha(X)$ is a constant such that the probability of a $X$ being greater than $v_\alpha(X)$ is $\beta$. Also observe that $\mathbb{P}(K_{n,\beta} = k) = \mathbb{P}(k$ from $\{X_i\}_{i=1}^n$ have values in

$[v_\alpha(X), b]$). Using the above two statements one can easily see that $K_{n,\beta}$ follows a binomial distribution with parameters $n$ and $\beta$.

Consider $k$ I.I.D. random variables $\{\tilde{X}_i\}_{i=1}^k$ which are distributed according to $\mathbb{P}(X \in \cdot \,|X \in [v_\alpha(X), b])$. By conditioning on $K_{n,\beta} = k$, one can observe using symmetry that $\frac{1}{k}\sum_{i=1}^k X_{[i]}$ and $\frac{1}{k}\sum_{i=1}^k \tilde{X}_i$ have the same distribution. We'll next bound the probability $\mathbb{P}(\hat{c}_{n,\alpha}(X) \le c_\alpha(X) - \varepsilon|K_{n,\beta} = k)$ for different values of $k$. Now,

$$\mathbb{P}(\hat{c}_{n,\alpha}(X) \le c_\alpha(X) - \varepsilon) = \sum_{k=0}^n \mathbb{P}(K_{n,\beta} = k)\mathbb{P}(A)$$

$$\le \underbrace{\sum_{k=0}^{\lfloor n\beta \rfloor} \mathbb{P}(K_{n,\beta} = k)\mathbb{P}(A)}_{I_2} + \underbrace{\sum_{k=\lceil n\beta \rceil}^n \mathbb{P}(K_{n,\beta} = k)\mathbb{P}(A)}_{I_1}$$

where $\mathbb{P}(A) = \mathbb{P}(\hat{c}_{n,\alpha}(X) \le c_\alpha(X) - \varepsilon|K_{n,\beta} = k)$.

**Bounding $I_1$**

Note that $k \ge \lceil n\beta \rceil$. We'll begin by bounding $P(A)$.

$$\mathbb{P}(\hat{c}_{n,\alpha}(X) \le c_\alpha(X) - \varepsilon|K_{n,\beta} = k)$$

$$\le \mathbb{P}\left(\frac{1}{\lceil n\beta \rceil} \sum_{i=1}^{\lceil n\beta \rceil} X_{[i]} \le c_\alpha(X) - \varepsilon|K_{n,\beta} = k\right) \text{ (using 4b)}$$

$$\le \mathbb{P}\left(\frac{1}{k} \sum_{i=1}^k X_{[i]} \le c_\alpha(X) - \varepsilon|K_{n,\beta} = k\right) \quad (\because f(\cdot) \text{ is decreasing})$$

$$= \mathbb{P}\left(\frac{1}{k} \sum_{i=1}^k \tilde{X}_i \le c_\alpha(X) - \varepsilon\right)$$

$$\le \exp\left(-\frac{k\varepsilon^2}{2\mathbb{E}[\tilde{X}^2] + 2b\varepsilon/3}\right) \text{ (using Bernstein's inequality)}$$

$\mathrm{Supp}(\tilde{X}) \in [v_\alpha(X), b]$. In worst case, $v_\alpha(X) = -b$. Therefore, $\mathrm{Supp}(\tilde{X}) \in [-b, b]$ and $\mathbb{E}[\tilde{X}^2] \le b^2$. Hence,

$$\mathbb{P}(\hat{c}_{n,\alpha}(X) \le c_\alpha(X) - \varepsilon|K_{n,\beta} = k) \le \exp\left(-\frac{k\varepsilon^2}{2b^2 + 2b\varepsilon/3}\right)$$

Hence, we have the following:

$$I_1 = \sum_{k=\lceil n\beta \rceil}^n \binom{n}{k}\beta^k(1-\beta)^{n-k}\mathbb{P}(\hat{c}_{n,\alpha}(X) \le c_\alpha(X) - \varepsilon|K_{n,\beta} = k)$$

$$\le \sum_{k=\lceil n\beta \rceil}^n \binom{n}{k}\left(\beta\exp\left(-\frac{\varepsilon^2}{2b^2 + 2b\varepsilon/3}\right)\right)^k (1-\beta)^{n-k}$$

$$\le \left(1 - \beta + \beta\exp\left(-\frac{\varepsilon^2}{2b^2 + 2b\varepsilon/3}\right)\right)^n$$

$$\le \exp\left(-\beta n\left(1 - \exp\left(-\frac{\varepsilon^2}{2b^2 + 2b\varepsilon/3}\right)\right)\right) \quad (\because e^x \ge 1 + x \;\forall x \in \mathbb{R})$$

Now, let's bound $1 - \exp\left(-\frac{\varepsilon^2}{2b^2 + 2b\varepsilon/3}\right)$. We know that $1 - e^{-x} \ge x - x^2/2 = x(1 - x/2)$. One can easily verify that $\frac{\varepsilon^2}{2b^2 + 2b\varepsilon/3}$ is an increasing function of $\varepsilon$ if $\varepsilon \ge 0$. Putting $\varepsilon = 2b$, we get,

$1 - \frac{1}{2}\frac{\varepsilon^2}{2b^2+2b\varepsilon/3} \geq 1 - \frac{3}{5} = \frac{2}{5}$. Hence, $1 - \exp\left(-\frac{\varepsilon^2}{2b^2+2b\varepsilon/3}\right) \geq \frac{2}{5}\frac{\varepsilon^2}{2b^2+2b\varepsilon/3}$. Therefore,

$$I_1 \leq \exp\left(-\beta n\left(\frac{(\varepsilon/b)^2}{5+5\varepsilon/(3b)}\right)\right)$$

**Bounding $I_2$**

Note that $k \leq \lfloor n\beta \rfloor$. We'll again start by bounding $\mathbb{P}(A)$.

$$\mathbb{P}(\hat{c}_{n,\alpha}(X) \leq c_\alpha(X) - \varepsilon | K_{n,\beta} = k) \leq \mathbb{P}\left(\frac{1}{n\beta}\sum_{i=1}^{\lfloor n\beta \rfloor} X_{[i]} \leq c_\alpha(X) - \varepsilon \middle| K_{n,\beta} = k\right) \text{ (Using 4a)}$$

$$\leq \mathbb{P}\left(\frac{1}{k}\sum_{i=1}^{k} X_{[i]} \leq \frac{n\beta}{k}(c_\alpha(X) - \varepsilon) \middle| K_{n,\beta} = k\right) \;(\because\; k \leq \lfloor n\beta \rfloor)$$

$$\leq \mathbb{P}\left(\frac{1}{k}\sum_{i=1}^{k} X_{[i]} \leq c_\alpha(X) + \left(\frac{n\beta}{k} - 1\right)b - \frac{n\beta\varepsilon}{k} \middle| K_{n,\beta} = k\right) \;(\because\; c_\alpha(X) \leq b)$$

**Case 1** $\varepsilon \in [b, 2b]$

Let $\varepsilon_1(k) = \frac{n\beta\varepsilon}{k} + \left(1 - \frac{n\beta}{k}\right)b = b\left(1 + \left(\frac{\varepsilon}{b} - 1\right)\frac{n\beta}{k}\right)$. Note that $\varepsilon_1(k) > 0$ for all $k$ as $\varepsilon \geq b$. Also note that $\varepsilon_1(k)$ decreases as $k$ increases. As $k \leq n\beta$, $\varepsilon_1(k) \geq \varepsilon$.

$$\mathbb{P}\left(\frac{1}{k}\sum_{i=1}^{k} X_{[i]} \leq c_\alpha(X) - \varepsilon_1(k) | K_{n,\beta} = k\right) = \mathbb{P}\left(\frac{1}{k}\sum_{i=1}^{k} \tilde{X}_i \leq c_\alpha(X) - \varepsilon_1(k)\right)$$

$$\leq \exp\left(-\frac{k\varepsilon_1^2(k)}{2b^2 + 2b\varepsilon_1(k)/3}\right)$$

$$\overset{(a)}{\leq} \exp\left(-\frac{k\varepsilon^2}{2b^2 + 2b\varepsilon/3}\right)$$

(a) above follows because $\frac{\varepsilon_1^2(k)}{2b^2+2b\varepsilon_1(k)/3}$ is an increasing function of $\varepsilon_1(k)$ and $\varepsilon_1(k) \geq \varepsilon$.

Using steps similar to that for bounding $I_1$, we have:

$$I_2 \leq \exp\left(-\beta n\left(\frac{(\varepsilon/b)^2}{5+5\varepsilon/(3b)}\right)\right); \quad \varepsilon \in [b, 2b]$$

**2.** $\varepsilon \in [0, b)$

Here, $\varepsilon_1(k) = \frac{n\beta\varepsilon}{k} - \left(\frac{n\beta}{k} - 1\right)b = b\left(1 - \left(1 - \frac{\varepsilon}{b}\right)\frac{n\beta}{k}\right)$. Note that $\varepsilon_1(k) > 0$ iff $k > n\beta(1 - \frac{\varepsilon}{b})$.

**Case 2.1** If $\varepsilon$ is very small such that $\lfloor n\beta \rfloor \leq n\beta\left(1 - \frac{\varepsilon}{b}\right)$, then $\varepsilon_1(k) \leq 0$. Let's bound $I_2$ for this case:

$$I_2 \leq \sum_{k=0}^{\lfloor n\beta \rfloor} \mathbb{P}(K_{n,\beta} = k)$$

$$= \mathbb{P}(K_{n,\beta} \leq \lfloor n\beta \rfloor)$$

$$\leq \mathbb{P}(K_{n,\beta} \leq n\beta(1 - \varepsilon/b))$$

$$\leq \exp\left(-n\beta\frac{\varepsilon^2}{2b^2}\right) \quad \text{(Chernoff on } K_{n,\beta}\text{)}$$

**Case 2.2** $n\beta(1 - \varepsilon/b) < \lfloor n\beta \rfloor$

Choose $k_\gamma^* = n\beta(1 - \gamma\varepsilon/b)$ for some $\gamma \in [0, 1]$. Then, $n\beta(1 - \varepsilon/b) \leq k_\gamma^* \leq n\beta$.

Assume $k_\gamma^* < \lfloor n\beta \rfloor$. The proof can can be easily adapted when $k_\gamma^* \geq \lfloor n\beta \rfloor$. As we will see, the bound on $I_2$ is looser when $k_\gamma^* < \lfloor n\beta \rfloor$.

For $k > k_\gamma^*$, $\varepsilon(k) > 0$. As $k$ increases, $\varepsilon_1(k)$ also increases.

Now, we'll bound $\mathbb{P}\Big(\frac{1}{k}\sum_{i=1}^{k} X_{[i]} \leq c_\alpha(X) - \varepsilon_1(k)\Big)$:

$$
\mathbb{P}\Big(\frac{1}{k}\sum_{i=1}^{k} X_{[i]} \leq c_\alpha(X) - \varepsilon_1(k)\Big) = \mathbb{P}\Big(\frac{1}{k}\sum_{i=1}^{k} \tilde{X}_i \leq c_\alpha(X) - \varepsilon_1(k)\Big)
$$

$$
\leq \begin{cases} \exp\Big(-\frac{k\varepsilon_1^2(k)}{2b^2 + 2b\varepsilon_1(k)/3}\Big); & k_\gamma^* < k \leq \lfloor n\beta \rfloor \\ 1; & k \leq k_\gamma^* \end{cases}
$$

$$
\overset{(a)}{\leq} \begin{cases} \exp\Big(k\frac{(1-\gamma)^2\varepsilon^2}{2(b-\gamma\varepsilon)^2 + 2(1-\gamma)\varepsilon(b-\gamma\varepsilon)/3}\Big); & k_\gamma^* < k \leq \lfloor n\beta \rfloor \\ 1; & k \leq k_\gamma^* \end{cases}
$$

$$
\overset{(b)}{\leq} \begin{cases} \exp\Big(-\frac{k(1-\gamma)^2(\varepsilon/b)^2}{2 + 2(1-\gamma)\varepsilon/(3b)}\Big); & k_\gamma^* < k \leq \lfloor n\beta \rfloor \\ 1; & k \leq k_\gamma^* \end{cases}
$$

(a) above follows because $\frac{\varepsilon_1^2(k)}{2b^2 + 2b\varepsilon_1(k)/3}$ is an increasing function of $\varepsilon_1(k)$ and $\varepsilon_1(k) \geq \frac{b(1-\gamma)\varepsilon}{b-\gamma\varepsilon}$.

(b) above follows because $b - \gamma\varepsilon \leq b$

Now, we'll bound $I_2$:

$$
I_2 \leq \sum_{k=0}^{\lfloor n\beta \rfloor} \mathbb{P}(K_{n,\beta} = k)\mathbb{P}\Big(\frac{1}{k}\sum_{i=1}^{k} \tilde{X}_i \leq c_\alpha(X) - \varepsilon_1(k)\Big)
$$

$$
\leq \underbrace{\sum_{k=0}^{\lfloor k_\gamma^* \rfloor} \binom{n}{k}\beta^k(1-\beta)^{n-k}}_{I_{2,a}} + \underbrace{\sum_{k=\lceil k_\gamma^* \rceil}^{\lfloor n\beta \rfloor} \binom{n}{k}\beta^k(1-\beta)^{n-k}\exp\Big(-\frac{k(1-\gamma)^2(\varepsilon/b)^2}{2 + 2(1-\gamma)\varepsilon/(3b)}\Big)}_{I_{2,b}}
$$

Let's bound $I_{2,a}$. This is very similar to *Case 2.1*.

$$
I_{2,a} = \sum_{k=0}^{\lfloor k_\gamma^* \rfloor} \binom{n}{k}\beta^k(1-\beta)^{n-k}
$$

$$
= \mathbb{P}\big(K_{n,\beta} \leq (1 - \gamma\varepsilon/b)n\beta\big)
$$

$$
\leq \exp\Big(-n\beta\frac{(\gamma\varepsilon)^2}{2b^2}\Big)
$$

If $\lceil k_\gamma^* \rceil > \lfloor n\beta \rfloor$, $I_{2,b} = 0$.

When $\lceil k_\gamma^* \rceil \leq \lfloor n\beta \rfloor$, let's bound $I_{2,b}$. This is very similar to bounding $I_1$.

$$
I_{2,b} \leq \Big(1 - \beta\Big(1 - \exp\Big(-\frac{(1-\gamma)^2(\varepsilon/b)^2}{2 + 2(1-\gamma)\varepsilon/(3b)}\Big)\Big)\Big)^n
$$

$$
\leq \exp\Big(-n\beta\Big(1 - \exp\Big(-\frac{(1-\gamma)^2(\varepsilon/b)^2}{2 + 2(1-\gamma)\varepsilon/(3b)}\Big)\Big)\Big)
$$

As $1 - \exp(-x) \geq x - x^2/2$ for $x \geq 0$

$$
1 - \exp\Big(-\frac{(1-\gamma)^2(\varepsilon/b)^2}{2 + 2(1-\gamma)\varepsilon/(3b)}\Big) \geq \frac{(1-\gamma)^2(\varepsilon/b)^2}{2 + 2(1-\gamma)\varepsilon/(3b)}\Big(1 - \frac{1}{2}\frac{(1-\gamma)^2(\varepsilon/b)^2}{2 + 2(1-\gamma)\varepsilon/(3b)}\Big)
$$

Further,

$$1 - \frac{1}{2}\frac{(1-\gamma)^2(\varepsilon/b)^2}{2 + 2(1-\gamma)\varepsilon/(3b)} \geq 1 - \frac{1}{2}\frac{(1-\gamma)^2}{2 + 2(1-\gamma)/3} \geq \frac{13}{16}$$

Hence, irrespective of whether $\lceil k_\gamma^* \rceil \leq \lfloor n\beta \rfloor$ or $\lceil k_\gamma^* \rceil > \lfloor n\beta \rfloor$:

$$I_{2,b} \leq \exp\left(-n\beta\left(\frac{13}{16}\frac{(1-\gamma)^2(\varepsilon/b)^2}{2 + 2(1-\gamma)\varepsilon/(3b)}\right)\right)$$

Now, we can bound $I_2$

$$I_2 \leq I_{2,a} + I_{2,b}$$

$$\leq \exp\left(-n\beta\frac{\gamma^2(\varepsilon/b)^2}{2}\right) + \exp\left(-n\beta\left(\frac{13}{16}\frac{(1-\gamma)^2(\varepsilon/b)^2}{2 + 2(1-\gamma)\varepsilon/(3b)}\right)\right)$$

Now, $\gamma^2 = \frac{13}{16}(1-\gamma)^2$ if $\gamma = 1/(1 + (16/13)^{0.5}) \approx 0.4740$. Put $\gamma = 0.4740$.

$$I_2 \leq 2\exp\left(-n\beta\frac{0.2247(\varepsilon/b)^2}{2 + 0.351\varepsilon/b}\right) = 2\exp\left(-n\beta\frac{(\varepsilon/b)^2}{8.9 + 1.561\varepsilon/b}\right)$$

Comparing this bound of $I_2$ with that of **Case 2.1**, it is not very difficult to see that the above bound is loose.

Comparing this bound of $I_2$ with that of **Case 1**, notice that the $8.9 + +1.561\varepsilon/b \geq 8.9$ whereas $5 + 5\varepsilon/3b \leq 8.34$. Hence, the above bound is the most general.

Finally, let's bound $I$:

$$I \leq I_1 + I_2$$

$$\leq \exp\left(-\beta n\left(\frac{(\varepsilon/b)^2}{5 + 5\varepsilon/(3b)}\right)\right) + 2\exp\left(-n\beta\frac{(\varepsilon/b)^2}{8.9 + 1.561\varepsilon/b}\right)$$

$$\leq 3\exp\left(-\beta n\frac{(\varepsilon/b)^2}{9 + 1.6\varepsilon/b}\right)$$

## A.2  Proof of 3b

Let's prove the second part of this theorem now which is the inequality 3b.

Again if $\varepsilon \geq 2b$,

$$\mathbb{P}(\hat{c}_{n,\alpha}(X) \geq c_\alpha(X) + \varepsilon) = 0$$

Hence, we're interested in the case where $\varepsilon \in [0, 2b)$. We'll again condition on random variable $K_{n,\beta}$. Remember that $K_{n,\beta}$ follows a binomial distribution with parameters $n$ and $\beta$.

The random variables $\{\tilde{X}_i\}_{i=1}^k$ are distributed according to $\mathbb{P}(X \in \cdot \,|\, X \in [v_\alpha(X), b])$. By conditioning of $K_{n,\beta} = k$ distributions of $\frac{1}{k}\sum_{i=1}^k X_{[i]}$ and $\frac{1}{k}\sum_{i=1}^k \tilde{X}_i$ are same by symmetry. The steps are very similar to that for proving 3a.

$$\mathbb{P}(\hat{c}_{n,\alpha}(X) \geq c_\alpha(X) + \varepsilon) = \sum_{k=0}^n \mathbb{P}(K_{n,\beta} = k)\mathbb{P}(A)$$

$$\leq \underbrace{\sum_{k=0}^{\lfloor n\beta \rfloor} \mathbb{P}(K_{n,\beta} = k)\mathbb{P}(A)}_{I_1} + \underbrace{\sum_{k=\lceil n\beta \rceil}^n \mathbb{P}(K_{n,\beta} = k)\mathbb{P}(A)}_{I_2}$$

where $\mathbb{P}(A) = \mathbb{P}(\hat{c}_{n,\alpha}(X) \geq c_\alpha(X) + \varepsilon | K_{n,\beta} = k)$. Notice that $I_1$ and $I_2$ got interchanged from A.1

**Bounding $I_1$**

Note that $k \leq \lfloor n\beta \rfloor$. Let's bound $\mathbb{P}(A)$ for this case:

$$\mathbb{P}(\hat{c}_{n,\alpha}(X) \geq c_\alpha(X) + \varepsilon | K_{n,\beta} = k) \leq \mathbb{P}\Big(\frac{1}{\lfloor n\beta \rfloor} \sum_{i=0}^{\lfloor n\beta \rfloor} X_{[i]} \geq c_\alpha(X) + \varepsilon | K_{n,\beta} = k\Big) \text{ (using 4b)}$$

$$\leq \mathbb{P}\Big(\frac{1}{k} \sum_{i=1}^{k} X_{[i]} \geq c_\alpha(X) + \varepsilon | K_{n,\beta} = k\Big) \text{ } (\because f(\cdot) \text{ is decreasing})$$

$$= \mathbb{P}\Big(\frac{1}{k} \sum_{i=1}^{k} \tilde{X}_i \geq c_\alpha(X) + \varepsilon\Big)$$

$$\leq \exp\Big(\frac{k(\varepsilon/b)^2}{2 + 2\varepsilon/(3b)}\Big) \text{ (Using Bernstein's Inequality)}$$

Let's bound $I_1$ now:

$$I_1 \leq \sum_{k=0}^{\lfloor n\beta \rfloor} \binom{n}{k} \Big(\beta\exp\Big(\frac{(\varepsilon/b)^2}{2 + 2\varepsilon/(3b)}\Big)\Big)^k (1-\beta)^{n-k}$$

$$\leq \Big(1 - \beta\Big(1 - \exp\Big(\frac{(\varepsilon/b)^2}{2 + 2\varepsilon/(3b)}\Big)\Big)\Big)^n$$

$$\leq \exp\Big(-n\beta\Big(1 - \exp\Big(\frac{(\varepsilon/b)^2}{2 + 2\varepsilon/(3b)}\Big)\Big)\Big) \text{ } (\because e^x \geq 1 + x)$$

$$\leq \exp\Big(-n\beta\frac{(\varepsilon/b)^2}{5 + 5\varepsilon/(3b)}\Big)$$

The last step is the same as that used for bounding $I_1$ in the previous proof.

**Bounding $I_2$:**

Note that $k \geq \lceil n\beta \rceil$. Let's begin by bounding $\mathbb{P}(A)$:

$$\mathbb{P}(\hat{c}_{n,\alpha}(X) \geq c_\alpha(X) + \varepsilon | K_{n,\beta} = k) \leq \mathbb{P}\Big(\frac{1}{n\beta} \sum_{i=1}^{\lceil n\beta \rceil} X_{[i]} \geq c_\alpha(X) + \varepsilon | K_{n,\beta} = k\Big) \text{ (using 4a)}$$

$$\leq \mathbb{P}\Big(\frac{1}{n\beta} \sum_{i=1}^{k} X_{[i]} \geq c_\alpha(X) + \varepsilon | K_{n,\beta} = k\Big) \text{ } (\because k \geq \lceil n\beta \rceil)$$

$$= \mathbb{P}\Big(\frac{1}{k} \sum_{i=1}^{k} X_{[i]} \geq \frac{n\beta}{k}(c_\alpha(X) + \varepsilon) | K_{n,\beta} = k\Big)$$

$$\leq \mathbb{P}\Big(\frac{1}{k} \sum_{i=1}^{k} X_{[i]} \geq c_\alpha(X) + \frac{n\beta\varepsilon}{k} - \Big(1 - \frac{n\beta}{k}\Big)b \Big| K_{n,\beta} = k\Big)$$

Let $\varepsilon_1(k) = \frac{n\beta\varepsilon}{k} - \Big(1 - \frac{n\beta}{k}\Big)b = b\Big((1 + \frac{\varepsilon}{b})\frac{n\beta}{k} - 1\Big)$. Notice that $\varepsilon_1(k) \geq 0$ if $k \leq (1 + \frac{\varepsilon}{b})n\beta$.

Unlike A.1, we can consider the entire range $\varepsilon \in [0, 2b]$.

**Case 1.1** If $\varepsilon$ is very small such that $(1 + \frac{\varepsilon}{b})n\beta \leq \lceil n\beta \rceil$, then $\varepsilon_1(k) \leq 0$. Note that $\frac{\varepsilon}{b} \leq 2$ and a suitable form of Chernoff bound will be used ahead.

Let's bound $I_2$ in this case:

$$
\begin{aligned}
I_2 &\leq \sum_{k=\lceil n\beta \rceil}^{n} \mathbb{P}(K_{n,\beta} = k) \\
&= \mathbb{P}(K_{n,\beta} \geq \lceil n\beta \rceil) \\
&\leq \mathbb{P}\Big(K_{n,\beta} \geq (1 + \varepsilon/b)n\beta\Big) = \lceil n\beta \rceil) \\
&\leq \exp\Big(-n\beta \frac{(\varepsilon/b)^2}{4}\Big) \text{ (Chernoff on } K_{n,\beta})
\end{aligned}
$$

**Case 1.2** $(1 + \frac{\varepsilon}{b})n\beta > \lceil n\beta \rceil$

We choose $k_\gamma^* = (1 + \frac{\gamma\varepsilon}{b})n\beta$ for some $\gamma \in [0,1]$. Note that $(1 + \frac{\varepsilon}{b})n\beta \geq k_\gamma^* \geq n\beta$. Assume that $k_\gamma^* > \lceil n\beta \rceil$. The proof when $k_\gamma^* \leq \lceil n\beta \rceil$ easily follows. We'll also see that the bound on $I_2$ is looser when $k_\gamma^* > \lceil n\beta \rceil$.

Note that $\varepsilon_1(k)$ decreases as $k$ increases. Now,

$$
\begin{aligned}
\mathbb{P}\Big(\frac{1}{k}\sum_{i=1}^{k} X_{[i]} \geq c_\alpha(X) + \varepsilon_1(k)\Big) &= \mathbb{P}\Big(\frac{1}{k}\sum_{i=1}^{k} \tilde{X}_i \geq c_\alpha(X) + \varepsilon_1(k) | K_{n,\beta} = k\Big) \\
&\leq \begin{cases} \exp\Big(-k\frac{(\varepsilon_1(k)/b)^2}{2 + 2\varepsilon_1(k)/(3b)}\Big); & \lceil n\beta \rceil \leq k < k_\gamma^* \\ 1; & k \geq k_\gamma^* \end{cases} \\
&\overset{(a)}{\leq} \begin{cases} \exp\Big(-k\frac{(1-\gamma)^2(\varepsilon/b)^2}{2(1+\gamma\varepsilon/b)^2 + 2(1+\gamma\varepsilon/b)(1-\gamma)\varepsilon/(3b)}\Big); & \lceil n\beta \rceil \leq k < k_\gamma^* \\ 1; & k \geq k_\gamma^* \end{cases} \\
&\overset{(b)}{\leq} \begin{cases} \exp\Big(-k\frac{(1-\gamma)^2(\varepsilon/b)^2}{2(1+2\gamma)^2 + 2(1+2\gamma)(1-\gamma)\varepsilon/(3b)}\Big); & \lceil n\beta \rceil \leq k < k_\gamma^* \\ 1; & k \geq k_\gamma^* \end{cases}
\end{aligned}
$$

(a) above follows because $\frac{(\varepsilon_1(k)/b)^2}{2 + 2\varepsilon_1(k)/(3b)}$ is an increasing function of $\varepsilon_1(k)$ and $\varepsilon_1(k)$ decreases as $k$ increases.

(b) above follows because $(1 + \gamma\varepsilon/b) \leq (1 + 2\gamma)$.

Now, we'll bound $I_2$:

$$
\begin{aligned}
I_2 &\leq \sum_{k=\lceil n\beta \rceil}^{n} \mathbb{P}(K_{n,\beta} = k)\mathbb{P}\Big(\frac{1}{k}\sum_{i=1}^{k} \tilde{X}_i \geq c_\alpha(X) + \varepsilon_1(k) | K_{n,\beta} = k\Big) \\
&\leq \underbrace{\sum_{k=\lceil k_\gamma^* \rceil}^{n} \binom{n}{k}\beta^k(1-\beta)^{n-k}}_{I_{2,a}} + \underbrace{\sum_{k=\lceil n\beta \rceil}^{\lfloor k_\gamma^* \rfloor} \binom{n}{k}\beta^k \exp\Big(-k\frac{(1-\gamma)^2(\varepsilon/b)^2}{8(1+2\gamma)^2 + 2(1+2\gamma)(1-\gamma)\varepsilon/(3b)}\Big)(1-\beta)^{n-k}}_{I_{2,b}}
\end{aligned}
$$

Let's bound $I_{2,a}$ first. This is very similar to *Case 1.1*. Here, $\frac{\gamma\varepsilon}{b} \leq 2$ and a suitable Chernoff bound is used ahead.

$$
\begin{aligned}
I_{2,a} &= \mathbb{P}(K_{n,\beta} \geq k_\gamma^*) \\
&\leq \mathbb{P}\Big(K_{n,\beta} \geq (1 + \gamma\varepsilon/b)n\beta\Big) \\
&\leq \exp\Big(-n\beta\frac{\gamma^2(\varepsilon/b)^2}{4}\Big) \text{ (Chernoff on } K_{n,\beta})
\end{aligned}
$$

If $\lfloor k_\gamma^* \rfloor < \lceil n\beta \rceil$, then $I_{2,b} = 0$. When $\lfloor k_\gamma^* \rfloor \geq \lceil n\beta \rceil$, let's bound $I_{2,b}$:

$$
\begin{aligned}
I_{2,b} &\leq \Big(1 - \beta\Big(1 - \exp\Big(-\frac{(1-\gamma)^2(\varepsilon/b)^2}{2(1+2\gamma)^2 + 2(1+2\gamma)(1-\gamma)\varepsilon/(3b)}\Big)\Big)\Big) \\
&\leq \exp\Big(-n\beta\Big(1 - \exp\Big(-\frac{(1-\gamma)^2(\varepsilon/b)^2}{2(1+2\gamma)^2 + 2(1+2\gamma)(1-\gamma)\varepsilon/(3b)}\Big)\Big)\Big)
\end{aligned}
$$

We know $1 - e^{-x} \geq x - x^2/2 = x(1 - x/2)$. Now,

$$g(\varepsilon, \gamma) = 1 - \frac{1}{2} \frac{(1-\gamma)^2(\varepsilon/b)^2}{2(1+2\gamma)^2 + 2(1+2\gamma)(1-\gamma)\varepsilon/(3b)}$$

$$\overset{(a)}{\geq} 1 - \frac{1}{2} \frac{2(1-\gamma)^2}{(1+2\gamma)^2 + 2(1+2\gamma)(1-\gamma)/3}$$

$$\overset{(b)}{\geq} \frac{2}{5}$$

(a) above follows because $g(\varepsilon, \gamma)$ decreases with $\varepsilon$. We put $\varepsilon = 2b$.

(b) above follows because increase in $\gamma$ increases the RHS of second step. Hence, we put $\gamma = 0$.
Irrespective of whether $\lfloor k_\gamma^* \rfloor < \lceil n\beta \rceil$ or $\lfloor k_\gamma^* \rfloor \geq \lceil n\beta \rceil$ :

$$I_{2,b} \leq \exp\left( - n\beta \frac{2}{5} \frac{(1-\gamma)^2(\varepsilon/b)^2}{2(1+2\gamma)^2 + 2(1+2\gamma)(1-\gamma)\varepsilon/(3b)} \right)$$

Bounding $I_2$ for this case, we get

$$I_2 \leq I_{2,a} + I_{2,b}$$

$$\leq \exp\left( - n\beta \frac{\gamma^2(\varepsilon/b)^2}{4} \right) + \exp\left( - n\beta \frac{2}{5} \frac{(1-\gamma)^2(\varepsilon/b)^2}{2(1+2\gamma)^2 + 2(1+2\gamma)(1-\gamma)\varepsilon/(3b)} \right)$$

Equate $\frac{\gamma^2}{4} = \frac{2}{5} \frac{(1-\gamma)^2}{2(1+2\gamma)^2}$ to get $\gamma = 0.3458546$.

$$I_2 \leq \exp\left( - 0.0299 n\beta(\varepsilon/b)^2 \right) + \exp\left( - n\beta \frac{0.0299(\varepsilon/b)^2}{1 + 0.1289\varepsilon/b} \right)$$

$$\leq 2\exp\left( - n\beta \frac{0.0299(\varepsilon/b)^2}{1 + 0.1289\varepsilon/b} \right)$$

$$= 2\exp\left( - n\beta \frac{(\varepsilon/b)^2}{33.44 + 4.311\varepsilon/b} \right)$$

The bound obtained on $I_2$ in **Case 1.1** is tighter than the above bound. But we need to take the looser bound because our bound should be valid for all $\varepsilon \in [0, 2b]$. Hence, we take the above bound on $I_2$.

Finally, we can bound $I$:

$$I \leq I_1 + I_2$$

$$\leq \exp\left( - n\beta \frac{(\varepsilon/b)^2}{5 + 5\varepsilon/(3b)} \right) + 2\exp\left( - n\beta \frac{(\varepsilon/b)^2}{33.44 + 4.311\varepsilon/b} \right)$$

$$\leq 3\exp\left( - n\beta \frac{(\varepsilon/b)^2}{34 + 4.4\varepsilon/b} \right)$$

# B   CVaR Concentration for Heavy Tailed Random Variables (Proof of Theorem 2)

We begin by bounding the bias in CVaR resulting from our truncation. It is important to note that so long as $b > |v_\alpha(X)|$, $v_\alpha(X) = v_\alpha(X^{(b)})$. Thus, for $b > |v_\alpha(X)|$,

$$|c_\alpha(X) - c_\alpha(X^{(b)})|$$

$$= c_\alpha(X) - c_\alpha(X^{(b)})$$

$$= \frac{1}{1-\alpha} \left( \mathbb{E}[X\mathbb{1}\{X \geq v_\alpha(X)\}] - \mathbb{E}[X^{(b)}\mathbb{1}\{X \geq v_\alpha(X)\}] \right)$$

$$= \frac{1}{1-\alpha} \mathbb{E}[X\mathbb{1}\{|X| > b\}\mathbb{1}\{X \geq v_\alpha(X)\}]$$

$$\overset{(a)}{=} \frac{1}{1-\alpha} \mathbb{E}[X\mathbb{1}\{X > b\}] \overset{(b)}{\leq} \frac{B}{(1-\alpha)b^{p-1}}. \tag{5}$$

Here, $(a)$ is a consequence of $b > |v_\alpha(X)|$. The bound $(b)$ follows from

$$\mathbb{E}[X\mathbb{1}\{X > b\}] \le \mathbb{E}\left[\frac{X^p}{X^{p-1}}\mathbb{1}\{X > b\}\right]$$

$$\le \frac{1}{b^{p-1}}\mathbb{E}\left[|X|^p\right] \le \frac{B}{b^{p-1}}.$$

It follows from (5) that for

$$b > \max\left(|v_\alpha(X)|, \left[\frac{2B}{\Delta(1-\alpha)}\right]^{\frac{1}{p-1}}, \frac{\Delta}{2}\right),$$

$|c_\alpha(X) - c_\alpha(X^{(b)})| \le \frac{\Delta}{2}$. Thus, for $b$ satisfying (2), we have

$$\Pr\left(|c_\alpha(X) - \hat{c}_{n,\alpha}^{(b)}(X)| \ge \Delta\right)$$

$$\le \Pr\left(|c_\alpha(X) - c_\alpha(X^{(b)})| + |c_\alpha(X^{(b)}) - \hat{c}_{n,\alpha}(X^{(b)})| \ge \Delta\right)$$

$$\overset{(a)}{\le} \Pr\left(|c_\alpha(X^{(b)}) - \hat{c}_{n,\alpha}(X^{(b)})| \ge \frac{\Delta}{2}\right)$$

$$\overset{(b)}{\le} 6\exp\left(-n(1-\alpha)\frac{(\Delta/b)^2}{4(34 + 4.4\Delta/(2b))}\right)$$

$$\overset{(c)}{\le} 6\exp\left(-n(1-\alpha)\frac{(\Delta/b)^2}{154}\right).$$

Here, $(a)$ follows the bound on $|c_\alpha(X) - c_\alpha(X^{(b)})|$ obtained earlier. To get $(b)$, we invoke Theorem 1. Finally, $(c)$ follows since $b > \Delta/2$. This completes the proof.

## C Error Bounds for Generalized Successive Rejects (Proof of Theorem 3 and Theorem 4)

The probability of error of the generalized successive rejects algorithm can be upper bounded in the following manner. During phase $k$, at least one of the $k$ worst arms is surviving. If the optimal arm $i^*$ is dismissed at the end of phase $k$, it means:

$$\xi_1\mu_{n_k}^\dagger(i^*) + \xi_2\hat{c}_{n_k,\alpha}^\dagger(i^*) \ge \min_{i \in \{(K),(K-1),\cdots,(K+1-k)\}} \xi_1\mu_{n_k}^\dagger[i] + \xi_2\hat{c}_{n_k,\alpha}^\dagger[i]$$

By using the union bound, we get:

$$p_e \le \sum_{k=1}^{K-1}\sum_{i=K+1-k}^{K} \mathbb{P}(\xi_1\mu_{n_k}^\dagger(i^*) + \xi_2\hat{c}_{n_k,\alpha}^\dagger(i^*) \ge \xi_1\mu_{n_k}^\dagger[i] + \xi_2\hat{c}_{n_k,\alpha}^\dagger[i])$$

$$= \sum_{k=1}^{K-1}\sum_{i=K+1-k}^{K} \mathbb{P}\big(\xi_1(\mu_{n_k}^\dagger(i^*) - \mu(i^*) - (\mu_{n^k}^\dagger[i] - \mu[i]))$$

$$+ \xi_2(\hat{c}_{n_k,\alpha}^\dagger(i^*) - c_\alpha(i^*) - (\hat{c}_{n_k,\alpha}^\dagger[i] - c_\alpha[i])) \ge \Delta[i]\big)$$

$$\le \sum_{k=1}^{K-1}\sum_{i=K+1-k}^{K} \mathbb{P}(\xi_1(\mu_{n_k}^\dagger(i^*) - \mu(i^*)) \ge \Delta[i]/4) + \mathbb{P}(\xi_1(\mu[i] - \mu_{n_k}^\dagger[i]) \ge \Delta[i]/4)$$

$$+ \mathbb{P}(\xi_2(\hat{c}_{n_k,\alpha}^\dagger(i^*) - c_\alpha(i^*)) \ge \Delta[i]/4) + \mathbb{P}(\xi_2(c_\alpha[i] - \hat{c}_{n_k,\alpha}^\dagger[i]) \ge \Delta[i]/4)$$

We've assumed that all the arms satisfy **C2**. For each arm $i$, we have high probability bounds for $|\mu_n^\dagger(i) - \mu(i)|$ and $|\hat{c}_{n,\alpha}^\dagger(i) - c_\alpha(i)|$ in terms of arm independent parameters $B$ and $p$, we can upper bound $p_e$ as follows:

$$p_e \le \sum_{k=1}^{K-1} k\big[\mathbb{P}(|\mu_{n_k}^\dagger(\cdot) - \mu(\cdot)| \ge \Delta[K+1-k]/(4\xi_1))$$

$$+ \mathbb{P}(|\hat{c}_{n_k,\alpha}^\dagger(\cdot) - c_\alpha(\cdot)| \ge \Delta[K+1-k]/(4\xi_2))\big]$$

By bounding $\mathbb{P}(|\mu_n^\dagger(\cdot) - \mu(\cdot)| \geq \Delta)$ and $\mathbb{P}(|\hat{c}_{n_k,\alpha}^\dagger(\cdot) - c_\alpha(\cdot)| \geq \Delta)$, we can bound $p_e$.

The statements of Theorem 3 and Theorem 4 follows easily from the following two lemmas:

**Lemma 2.** *By setting the truncation parameter as $b = n^q$ where $q > 0$,*

$$\mathbb{P}(|\mu(k) - \hat{\mu}_n^\dagger(k)| \geq \Delta) \leq 2exp\Big(-n^{1-q}\frac{\Delta}{4}\Big) \text{ for } n > n^*, \text{ where}$$

$$n^* = \Big(\frac{3B}{\Delta}\Big)^{\frac{1}{q\min(1,p-1)}}$$

**Lemma 3.** *By setting the truncation parameter as $b = n^q$ where $q > 0$,*

$$\mathbb{P}(|c_\alpha(X) - \hat{c}_{n,\alpha}^\dagger(X)| \geq \Delta) \leq 6exp\Big(-n^{1-2q}\frac{\beta\Delta^2}{154}\Big) \text{ for } n > n^*, \text{ where}$$

$$n^* = \max\left(\Big(\frac{2B}{\beta\Delta}\Big)^{\frac{1}{q(p-1)}}, \Big(\frac{B}{\min(\alpha,\beta)}\Big)^{\frac{1}{qp}}, \Big(\frac{\Delta}{2}\Big)^{\frac{1}{q}}\right)$$

### C.1 Proof of Lemma 2

We'll use the following lemma to prove results for mean minimization

**Lemma 4.** *Assume that $\{X_i\}_{i=1}^n$ be $n$ I.I.D. samples drawn from the distribution of $X$ which satisfies condition **C2**, then with probability at least $1 - \delta$,*

$$|\mu(k) - \hat{\mu}_n^\dagger(k)| \leq \begin{cases} \frac{\sum_{i=1}^n B/b_i^{p-1}}{n} + \frac{2b_n\log(2/\delta)}{n} + \frac{B}{2b_n^{p-1}}; & p \in (1,2] \\ \frac{\sum_{i=1}^n B/b_i^{p-1}}{n} + \frac{2b_n\log(2/\delta)}{n} + \frac{B^{2/p}}{2b_n}; & p \in (2,\infty) \end{cases}$$

It is adapted from proof of Lemma 1 in Yu et al. [2018].

**Case 1** $p \in (1,2]$ Using Lemma 4, if $p \in (1,2]$:

$$|\mu(k) - \hat{\mu}_n^\dagger(k)| \leq \frac{\sum_{i=1}^n B/b_i^{p-1}}{n} + \frac{2b_n\log(2/\delta)}{n} + \frac{B}{2b_n^{p-1}}$$

$$\leq \frac{3B}{2n^{q(p-1)}} + \frac{2}{n^{1-q}}\log(2/\delta)$$

We want to find $n^*$ such that for all $n > n^*$:

$$\underbrace{\frac{3B}{2n^{q(p-1)}}}_{T_1} + \underbrace{\frac{2}{n^{1-q}}\log(2/\delta)}_{T_2} < \Delta$$

Sufficient condition to ensure the above inequality is to make the $T_1 < \Delta/2$ and $T_2 \leq \Delta/2$.

$T_1 \leq \Delta/2$ if:

$$n > \Big(\frac{3B}{\Delta}\Big)^{\frac{1}{q(p-1)}}$$

Equating $T_2 = \Delta/2$, we get:

$$\delta = 2\exp\Big(-n^{1-q}\frac{\Delta}{4}\Big)$$

**Case 2** $p \in (2,\infty)$

Using Lemma 4, if $p \in (2,\infty)$:

$$|\mu(k) - \hat{\mu}_n^\dagger(k)| \leq \frac{\sum_{i=1}^n B/b_i^{p-1}}{n} + \frac{2b_n\log(2/\delta)}{n} + \frac{B^{2/p}}{2b_n}$$

$$\leq \frac{B}{n^{q(p-1)}} + \frac{B}{2n^q} + \frac{2\log(2/\delta)}{n^{1-q}}$$

$$\leq \frac{3B}{2n^q} + \frac{2\log(2/\delta)}{n^{1-q}}$$

We want to find $n^*$ such that for all $n > n^*$:

$$\underbrace{\frac{3B}{2n^q}}_{T_1} + \underbrace{\frac{2\log(2/\delta)}{n^{1-q}}}_{T_2} < \Delta$$

Sufficient condition to ensure the above inequality is to make the $T_1 < \Delta/2$ and $T_2 \leq \Delta/2$.
$T_1 < \Delta/2$ if:

$$n > \left(\frac{3B}{\Delta}\right)^{\frac{1}{q}}$$

Equating $T_2 = \Delta/2$, we get:

$$\delta = 2\exp\left(-n^{1-q}\frac{\Delta}{4}\right)$$

## C.2 Bounding Magnitude of VaR

Before we prove Lemma 3, we'll first bound $|v_\alpha(X)|$ in terms of $B$, $p$ and $\alpha$.

**Lemma 5.**

$$|v_\alpha(X)| \leq \left(\frac{B}{min(\alpha, \beta)}\right)^{\frac{1}{p}}$$

*Proof.* If $v_\alpha(X) > 0$, by definition:

$$
\begin{aligned}
1 - \alpha &= \int_{v_\alpha(X)}^{\infty} dF_X(x) \\
&= \int_{v_\alpha(X)}^{\infty} |x|^p/|x|^p dF_X(x) \\
&\leq B/|v_\alpha(X)|^p
\end{aligned}
$$

Hence, $|v_\alpha(X)| \leq (\frac{B}{\beta})^{\frac{1}{p}}$.

If $v_\alpha(X) < 0$, by definition:

$$
\begin{aligned}
\alpha &= \int_{-\infty}^{v_\alpha(X)} dF_X(x) \\
&= \int_{-\infty}^{v_\alpha(X)} |x|^p/|x|^p dF_X(x) \\
&\leq B/|v_\alpha(X)|^p
\end{aligned}
$$

Hence, $|v_\alpha(X)| \leq (\frac{B}{\alpha})^{\frac{1}{p}}$. $\qquad\square$

## C.3 Proof of Lemma 3

The proof follows from Theorem 2 and Lemma 5. We're growing our truncation parameter as $n^q$.
Therefore,

$$b = n^q > \max\left(\frac{\Delta}{2}, \left(\frac{B}{\min(\alpha, \beta)}\right)^{\frac{1}{p}}, \left[\frac{2B}{\Delta(1-\alpha)}\right]^{\frac{1}{p-1}}\right)$$

# D  Error Bounds for Non-oblivious Algorithms

In the non-oblivious setting, error bounds for the generalized successive rejects algorithm follow
from the following two lemmas.

**Lemma 6.** *By setting the truncation parameter* $b > \left(\frac{3B}{\Delta}\right)^{\frac{1}{\min(1,p-1)}}$,

$$\mathbb{P}(|\mu(k) - \hat{\mu}_n^\dagger(k)| \geq \Delta) \leq 2exp\left(-n\frac{\Delta}{4b}\right).$$

**Lemma 7.** *By setting the truncation parameter* $b > \max\left(\frac{\Delta}{2}, \left(\frac{B}{\min(\alpha,\beta)}\right)^{\frac{1}{p}}, \left[\frac{2B}{\Delta(1-\alpha)}\right]^{\frac{1}{p-1}}\right)$,

$$\mathsf{Pr}\left(|c_\alpha(k) - \hat{c}_{n,\alpha}^{(b)}(k)| \geq \Delta\right) \leq 6exp\left(-n(1-\alpha)\frac{\Delta^2}{154b^2}\right).$$

Note that the truncation parameters here are not a function of $n$ and therefore we get an exponentially decaying bound.

## D.1 Proof of Lemma 6

**Lemma 8.** *By setting the truncation parameter* $b > \left(\frac{3B}{\Delta}\right)^{\frac{1}{\min(1,p-1)}}$ *where* $q > 0$,

$$\mathbb{P}(|\mu(k) - \hat{\mu}_n^\dagger(k)| \geq \Delta) \leq 2exp\left(-n\frac{\Delta}{4b}\right)$$

*Proof.* Using Lemma 4, by fixing the truncation parameter as $b$, and making simplifications, with probability $1 - \delta$, we have:

$$|\mu(k) - \hat{\mu}_n^\dagger(k)| \leq \begin{cases} \frac{3B}{2b^{p-1}} + \frac{2b\log(2/\delta)}{n}; & p \in (1,2] \\ \frac{3B}{2b} + \frac{2b\log(2/\delta)}{n}; & p \in (2,\infty) \end{cases}$$

**Case 1** $p \in (1,2]$ We're interested to find $b$ and $\delta$ such that for all values of $n$:

$$\underbrace{\frac{3B}{2b^{p-1}}}_{T_1} + \underbrace{\frac{2b\log(2/\delta)}{n}}_{T_2} < \Delta$$

A sufficient condition for the above equation to be valid is $T_1 < \Delta/2$ and $T_2 = \Delta/2$.

To ensure $T_1 < \Delta/2$, take $b > \left(\frac{3B}{\Delta}\right)^{\frac{1}{p-1}}$.

By equating $T_2 = \Delta/2$, we get $\delta = 2exp\left(-n\frac{\Delta}{4b}\right)$ where $b$ is what we found above.

**Case 2** $p \in (2,\infty)$ We're interested to find $b$ and $\delta$ such that for all values of $n$:

$$\underbrace{\frac{3B}{2b}}_{T_1} + \underbrace{\frac{2b\log(2/\delta)}{n}}_{T_2} < \Delta$$

A sufficient condition for the above equation to be valid is $T_1 < \Delta/2$ and $T_2 = \Delta/2$.

To ensure $T_1 = \Delta/2$, take $b > \frac{3B}{\Delta}$.

By equating $T_2 = \Delta/2$, we get $\delta = 2exp\left(-n\frac{\Delta}{4b}\right)$ where $b$ is what we found above. $\square$

## D.2 Proof of Lemma 7

Lemma 7 follows from Theorem 2 and Lemma 5.