[Reviews · NeurIPS 2019]

Reviewer 1



The main idea of this submission is the new concentration inequality, Theorem 2, proved for analyzing the proposed algorithm. To generalize from bounded random variable X to X that satisfies the mild assumptions made in the submission, the application of a truncation-based estimator plays a crucial role for developing a new concentration inequality. Another idea is to introduce the linear combination of the mean and CVaR studied in financial community as the optimization objective. This formulation includes the expected reward maximization and the CVaR minimization as special cases. The proposed Algorithm 1, Generalized successive rejects, identifies the arm with minimum loss and CVaR has bounded misidentification probability shown in Theorem 4. The resulting bounds are discussed and compared to that of non-oblivious settings, including the probability in terms of T (number of pulls), the minimum requirement of T, and the interaction between T and the truncation parameters. The authors also show that when the same information is provided as the non-oblivious setting, Algorithm 1 achieves the same performance as that proposed in the non-oblivious setting. Simulations verifying the performance of the proposed algorithm and demonstrating how truncation parameters affect the performance are provided. The reviewer has concerns about the submission. Please see Improvements for details. Originality. The submission relaxes the assumptions on the distributions. A novel concentration inequality is proved. Quality. The analysis is rigorous. The performance of this work is compared and discussed with the non-oblivious ones. Clarity. The submission and the analysis is well written. Significance. Bounds on the misidentification probabilities under milder assumptions on distributions are proved. A new concentration inequality is proved and could be applied to other research. ------------------------------ Update Thank you for the response. I am satisfied with the authors' reply, especially the first point clarifying the algorithm actually doesn't need C2.

Reviewer 2



The paper considers a new pure exploration problem where the target criterion is a linear combination of the mean and the conditional value at risk. While the algorithmic framework is the usual Successive Rejects (the fixed budget setting), the newly proposed estimator and its concentration inequality enables the algorithm to be agnostic (=oblivious) to the type of the distributions (up to very mild assumptions). The empirical results are compelling, and it suggests that the theory might be tightened up. Overall, a well-written paper. The key novelty is in the new estimator and concentration inequality that is not restricted to bounded random variables. The idea of setting b value as a function of n and enjoying guarantees when n is large enough seem like an interesting idea though similar concepts appear in the literature. I think the paper is in a good shape. It took me some time to appreciate "where" the distribution-obliviousness comes from, since the SR algorithm has been distribution-obliviousness, I thought (e.g., we don't need to know the variance when the arms are sub-Gaussian). However, I realized that it becomes interesting in the heavy-tailed distribution / CVaR regime where the estimator itself would require some knowledge on the distributions itself. I don't think the authors said anything incorrect, but maybe they can avoid confusion by adding more explanation. - "MAB formulations consider reward distributions with bounded support, typically [0, 1]. Moreover, the support is assumed to be known beforehand, and this knowledge is baked into the algorithm. " attacking this is quite weak, since many times you can relax this to \sigma^2-sub-Gaussianity, and for SR you don't need to know \sigma ahead of time. It'd be nice to address this case as well. - It might be helpful to explain that the sample average in used in the standard SR does not serve the purpose in heavy-tailed distribution / CVaR problems anymore, and naïve applications of existing techniques would require information about the distribution." # minor comments - Note the authors are not using the correct formatting for the submission, so there is no line numbers in the submission. This makes things a bit difficult. - I am pretty sure the definition of v_\alpha(X) should have ">= \alpha" rather than "<= \alpha". Otherwise v_\alpha(X) would be -\infty. - It would've been easier to define CVaR as E[Y | Y \ge VaR_\alpha(Y)] then say it becomes c_\alpha(X) defined in the paper since that's easier to understand the meaning. I myself had to track down the reference to dig it out. - In section 4, first paragraph, I think (\xi_1, \xi_2) = (1,0) for the first appearance rather than (0,1). - Figure 2 (a), b_m must be q_m? ==== [after rebuttal] I stand by my score. It'd be great if you could make the paper more accessible to nonexperts by adding definitions/explanations/references on the basic concept such as CVaR.

Reviewer 3



This paper proposes an algorithm for the multi-armed bandit, where the goal is to identify the arm that has the minimum value of the conditional value at risk (CVaR) of the reward, which may be heavy-tailed. The proposed algorithm is enabled by the new truncation-based estimator for the CVaR, for which new concentration bound is established. The new truncation-based estimator is of interest in its own right and may find applications beyond multi-armed bandit. Overall, the paper makes solid contributions primarily from mathematical perspectives. The proposed algorithm has limited applicability for the following reasons (please explain if there are misunderstandings): - The total number of arm-pulls needs to be known in advance to determine the number of arm-pulls in each stage. - To compute the estimator, all of the observed reward needs to be stored. I also wonder what real applications involve heavy-tailed reward. I would like to see experimental comparisons of the new truncation-based estimator against existing estimators of CVaR. --- I have read the responses from the authors. I am not completely sure exactly how the "doubling trick" and "summary statistics" can be used. I hope the authors explain them in more detail in the final version. Also, I understand that heavy-tailed distributions are ubiquitous but not sure exactly what applications would benefit from optimization via bandit. In any case, this paper makes sufficient contributions to be accepted.

[Author Response · NeurIPS 2019]

Response to comments of Reviewer 1:

**1. C1** is not essential to our analysis. All our technical statements hold even without **C1**; this assumption simply makes
the VaR and CVaR easier to express. On the other hand, **C2** (finiteness $(1 + \epsilon)$th moment for some $\epsilon > 0$) is essential to
our analysis, though it is only slightly more restrictive than assuming that the MAB problem is well- posed (which
corresponds to $\epsilon = 0$). Note that our algorithms do not know this $\epsilon$.

**2.** It turns out that the regret minimization framework is much harder to work within a distribution oblivious setting.
Indeed, to the best of our knowledge, none of the current algorithms for regret analysis are distribution oblivious.

**3.** We agree that it makes sense to include a brief review of various risk-aware statistical approaches in the literature –
we will do so if the paper is accepted. However, we think the Bayesian risk framework may not be directly suited to our
distribution oblivious setting, since it involves assigning a prior distribution over the unknown parameter space.

**4.** Linear combinations of the mean and a risk metric are analytically tractable, and therefore ubiquitous in the literature
(for example, in modern portfolio theory). Of course, linear combinations also come up naturally in Lagrange relaxations
of constrained optimization, such as minimizing expected loss subject to a given risk constraint.

**5.** Note that we only provide *upper bounds* on the misidentification probability in oblivious and non-oblivious settings.
We observe that the actual performance is often better than the bounds, as depicted in the figure.

Response to comments of Reviewer 2:

The reviewer correctly notes that SR-type algorithms are distribution oblivious with standard empirical estimators for
mean/CVaR, though these perform very poorly if reward distributions are heavy-tailed. This can be formalized via
*lower bounds* for these algorithms. If accepted, we will elaborate on this aspect in the final version of this paper. We
apologize for omitting the line numbers. Thank you for pointing out the typos; these will be fixed.

Response to comments of Reviewer 3:

**1.** Note that any SR-type algorithm requires apriori information about the time horizon. However, our uniform
exploration (UE) algorithm can be made *any-time* using the doubling trick.

**2.** We submit that the proposed algorithms do not require all rewards to be stored. For SR, since the truncation
parameters in each phase are known ahead of time, summary statistics of prior pulls can be stored for future rounds,
leading to an $O(K^2)$ memory requirement (rather than $O(T)$). For UE, $O(K)$ memory suffices, even for the any-time
modification.

**3.** On applicability: Heavy tailed distributions are ubiquitous, and are used to model financial rewards, losses due to
natural calamities, wealth, Internet file sizes, social media followers, etc. Our formulations can, therefore, be applied to
online learning problems in networking, economics, and finance. Of course, our CVaR concentration inequalities are of
independent value, even beyond the MAB framework.

**4.** On experimental comparisons between the conventional CVaR estimator and the oblivious truncation-based CVaR
estimator: Consider two pure exploration CVaR minimization scenarios. (a) **Light-tailed setting:** We have two
exponentially distributed arms with the means such that the arms have CVaRs 1.0 and 0.98 at confidence 0.95. (b)
**Heavy-tailed setting:** We have two Lomax distributed arms with shape parameter 2 and scale parameter set so that the
arms have CVaRs 1.0 and 0.9 at confidence 0.95. For each value of $T$, we run the SR algorithm 50000 times. In the
light-tailed setting (see Fig. 1a), the conventional empirical estimator works well, and truncation-aided bias-variance
tradeoff does not help. But in the heavy-tailed setting (see Fig. 1b), the conventional empirical estimator performs
poorly because of its considerable variance.

(a) Light Tailed Losses      (b) Heavy Tailed Losses

Figure 1: Comparison of conventional and truncation-based CVaR estimators

[Meta-Review · NeurIPS 2019]

There is a consensus among reviewers that this is a good paper worth acceptance. A few typos spotted by the area chair: - There is a typo in the definition of VaR: one of the inequalities should be flipped, otherwise \xi is trivially minus infinity. - There is an issue with the way Bubeck and Cesa-Bianchi (2012) appears in the references - "et al." should not be there.